# Potential, Limitations and Risks of Cannabis-Derived Products in Cancer Treatment

**DOI:** 10.3390/cancers15072119

**Published:** 2023-04-01

**Authors:** Herman J. Woerdenbag, Peter Olinga, Ellen A. Kok, Donald A. P. Brugman, Ulrike F. van Ark, Arwin S. Ramcharan, Paul W. Lebbink, Frederik J. H. Hoogwater, Daan G. Knapen, Derk Jan A. de Groot, Maarten W. Nijkamp

**Affiliations:** 1Department of Pharmaceutical Technology and Biopharmacy, University of Groningen, Antonius Deusinglaan 1, 9713 AV Groningen, The Netherlands; p.olinga@rug.nl (P.O.); ellen.kokx@hotmail.com (E.A.K.); donald.brugman@gmail.com (D.A.P.B.); ulrikevanark@gmail.com (U.F.v.A.); 2Transvaal Apotheek, Kempstraat 113, 2572 GC Den Haag, The Netherlands; a.ramcharan@transvaalapotheek.nl (A.S.R.); p.lebbink@transvaalapotheek.nl (P.W.L.); 3Department of Surgery, University Medical Center Groningen, Hanzeplein 1, 9713 GZ Groningen, The Netherlands; f.j.h.hoogwater@umcg.nl (F.J.H.H.); m.w.nijkamp@umcg.nl (M.W.N.); 4Department of Medical Oncology, University Medical Center Groningen, Hanzeplein 1, 9713 GZ Groningen, The Netherlands; d.g.knapen@umcg.nl (D.G.K.); d.j.a.de.groot@umcg.nl (D.J.A.d.G.)

**Keywords:** cancer treatment, cannabidiol (CBD), cannabinoids, cannabis products, delta-9-tetrahydrocannabinol (THC), drug interactions, oncology, quality of life, symptom management

## Abstract

**Simple Summary:**

It is easy to find success stories on the internet of patients with cancer who seem to benefit from using cannabis products. However, scientific substantiation is usually lacking. Therefore, there are critical questions among clinicians and other healthcare providers about the potential of cannabis products in cancer care. In this article, we aim to give direction for making choices about the responsible use of cannabis products in oncology by addressing the following questions: How does cannabis work? What is medicinal cannabis? What kind of cannabis products are in use? What is their legal status? Is there evidence for therapeutic effects in patients with cancer? What is the risk–benefit balance in terms of adverse effects, (potential) drug interactions, symptom management and antitumour activity? May cannabis products provide added value in the treatment of patients with cancer? We end up with an outlook and perspective determining the place of cannabis products in oncology.

**Abstract:**

The application of cannabis products in oncology receives interest, especially from patients. Despite the plethora of research data available, the added value in curative or palliative cancer care and the possible risks involved are insufficiently proven and therefore a matter of debate. We aim to give a recommendation on the position of cannabis products in clinical oncology by assessing recent literature. Various types of cannabis products, characteristics, quality and pharmacology are discussed. Standardisation is essential for reliable and reproducible quality. The oromucosal/sublingual route of administration is preferred over inhalation and drinking tea. Cannabinoids may inhibit efflux transporters and drug-metabolising enzymes, possibly inducing pharmacokinetic interactions with anticancer drugs being substrates for these proteins. This may enhance the cytostatic effect and/or drug-related adverse effects. Reversely, it may enable dose reduction. Similar interactions are likely with drugs used for symptom management treating pain, nausea, vomiting and anorexia. Cannabis products are usually well tolerated and may improve the quality of life of patients with cancer (although not unambiguously proven). The combination with immunotherapy seems undesirable because of the immunosuppressive action of cannabinoids. Further clinical research is warranted to scientifically support (refraining from) using cannabis products in patients with cancer.

## 1. Introduction 

The application of cannabis-derived products (hereinafter referred to as ‘cannabis products’) is a current topic of interest, especially among patients with cancer who lack perspectives for (further) treatment. The number of patients with cancer using cannabis products, both in a curative and palliative setting, is rising. Not-scientifically substantiated success stories, written by patients with cancer stating that they benefit from cannabis, can be easily found on the internet. This leaves clinicians with the question of whether to accept or oppose the use of cannabis products by the patients they treat. 

Despite the plethora of research data available in the literature, an evidence-based scientific background that underpins the clinical potential and possible risks of cannabis products in oncology is scarce. This provokes scepticism among healthcare professionals regarding the use, clinical safety and efficacy of cannabis products. Therefore, the advice often given to patients treated for cancer is to refrain from cannabis use. It is questionable whether this is the right attitude. Hesitance in prescribing medicinal cannabis products may encourage patients to seek salvation in non-medicinal-grade cannabis products, with all associated risks. In addition, in the palliative care of patients with cancer that are out-of-treatment, it remains valuable to explore all possibilities to improve the quality of life. 

This article is a narrative review that aims to give a recommendation on the position of cannabis products in clinical oncology, based on a critical assessment of recent literature (2016–2022) with connections to older scientific work. The literature review was conducted by searching PubMed and Web of Science and via a Google search. The search terms used included cannabis (products), combined with oncology, cancer, symptom management, quality of life and safety. The focus of the article is on the characteristics and quality of cannabis products and their impact on responsible use, on the potential of cannabis products in symptom management in palliative care, on the therapeutic application of cytostatic properties of cannabis products, and on the possible interaction of cannabis products with anticancer drugs and concurrent medication in patients with cancer and its translation to clinical pharmacotherapy. We end up with an outlook and perspective determining the place of cannabis products in oncology based on considerations around benefits and risks.

## 2. Pharmacognostic and Pharmacological Background

### 2.1. Cannabis Plants as a Source of Cannabinoids

Cannabis plants, commonly known as hemp, have been cultivated worldwide for centuries to serve industrial, recreational and medical purposes. In all cases, the botanical source is *Cannabis sativa* L., a member of the Cannabinaceae plant family. *C. sativa* is considered as the single species of the genus *Cannabis* although 35 synonyms are known. Various subspecies, varieties (chemotypes) and cultivars of *C. sativa* are distinguished, being strong determinants for their use [1,2,3]. 

For the biological activity of cannabis, the cannabinoids are the most important constituents [4]. They are also referred to as phytocannabinoids, to discriminate them from endocannabinoids (endogenic neurotransmitters that bind to cannabinoid receptors) and synthetic cannabinoids [5]. While considerable differences are seen regarding the qualitative and quantitative cannabinoid composition among varieties of plants, thus far 130 different structures have been elucidated as recently reviewed [2]. The highest contents are found in the resin and in the glandular trichomes of the flowering tops of female plants. The principal and most investigated cannabinoids are delta-9-tetrahydrocannabinol (THC) and cannabidiol (CBD) (Figure 1). When using cannabis products in the treatment of patients with cancer, the focus is nearly always on (one of) these two compounds [6,7,8,9,10,11,12,13]. 

### 2.2. Pharmacology of Cannabinoids

The pharmacological basis of cannabinoids mainly resides in their interaction with the endocannabinoid system (ECS) in the human body. The term ‘cannabinoid’ is used and defined as a compound with a THC-like structure, acting on cannabinoid receptors (according to the International Union of Basic & Clinical Pharmacology (IUPHAR)) [14]. The ECS consists of two G-protein-coupled endocannabinoid receptors, CB1 and CB2, for which endocannabinoids (arachidonic acid derivatives, mainly N-arachidonoylethanolamine (anandamide, AEA) and 2-arachidonoylglycerol (2-AG)) serve as full or partial agonists [5,15]. CB1 receptors are primarily located in the central nervous system and in non-neural peripheral tissues, while CB2 receptors are predominantly found in cells of the immune system and in the spleen [10,11]. 

The mechanism of action of THC and CBD has been profoundly investigated. THC is an agonist for both the CB1 and CB2 receptors, showing the highest affinity for the CB1 receptor. CBD acts as an antagonist for the CB1 and an inverse agonist for the CB2 receptor. Activation of the cannabinoid receptors results in the inhibition of adenylyl cyclase, decreasing cAMP [10,16] and thereby modulating ion channels [17]. For an elaborate explanation of the ECS and the signalling pathways involved, we refer to the reviews by Zou et al. [15] and Moreno et al. [18].

THC and CBD are also ligands for targeting transient receptor potential (TRP) channels, including TRPV1, TRPV2, TRPV3, TRPV4, TRPA1 and TRPM8. Cannabinoids have further been found to show an affinity for orphan G-protein-coupled receptors, such as GPR55. THC also interacts with receptors that are not part of the ECS, such as the PPARy nuclear receptor. Through the CB2 receptor, CBD may inhibit the expression of TNF-α, iNOS and COX-2, involved in inflammatory signalling pathways. CBD, after interaction with the CB2 receptor, inhibits adenosine uptake via the A2A receptor, thereby inhibiting inflammatory reactions. CBD inhibits the breakdown of anandamide, thereby counteracting the effect of THC as a cannabinoid receptor agonist. Finally, THC and CBD may affect and inhibit a number of transporters and enzymes involved in the distribution and metabolism of drug compounds [9,19,20,21,22,23,24,25,26].

Interesting from an oncology viewpoint is that it has been shown that the cannabinoid receptors play a role in the regulation of cell proliferation and cell death by acting on various signalling pathways [10,27], such as the regulation of phosphorylation and activation of mitogen-activated-protein kinases (MAPKs) and p38 MAPKs [28].

Thus, the overall effects of THC and CBD are brought about by a complex combination of mechanisms. While THC has affinity preferably for the CB1 receptor, resulting in its psychoactive effect, activation of CB2 via CBD results in immune modulation [29]. CBD is not psychoactive but may influence the psychoactive effects elicited by THC [30,31,32].

The effects of cannabis products are mainly based on the amount and ratio of THC and CBD, but other plant constituents present in cannabis products, such as terpenes and flavonoids, are likely to contribute to the biological activity as well. This phenomenon is called the ‘entourage effect’: the combination of different cannabinoids or cannabinoids and terpenes may enhance the overall activity [10,33]. For instance, in ER+/PR+, HER2+ and triple-negative breast cancer (both in vitro and in vivo), pure THC was found to be less potent than a botanical drug preparation [34], and the CB1-dependent behaviour in mice of the synthetic cannabinoid WIN55,212-2 was improved by co-administering cannabis terpenes (α-humulene and β-pinene) [35]. Cannflavin A, a cytotoxic flavonoid from the cannabis plant, has been shown to act synergistically with cannabinoids in two translational bladder cancer cell lines [25]. At present, there is no satisfactory proof for the existence of such an ‘entourage effect’ in humans. Such proof may be obtained from a solid RCT in which a full-spectrum cannabis product is compared with the equivalent pure-THC product.

Synthetic cannabinoids have been and are being developed as potential new drug entities [36,37], but these are beyond the scope of this article. Our focus of discussion is on THC and CBD, as major constituents of cannabis products.

## 3. Cannabis Products

### 3.1. Regulation of Cannabis Products

Because of the psychotropic action of THC, the production and supply of cannabis plants and products are submitted to narcotics regulations in many countries [4,38,39,40,41] with considerable variety regarding the legislative status. A complete ban on cannabis use, apparently focusing on the risks of abuse only, definitely ignores the medical potential. Proper and transparent regulation is needed for decision making by medical doctors and pharmacists regarding the responsible and therapeutic use of medicinal cannabis and cannabis products. Limited or unclear legislation will impair further and necessary scientific research on cannabis. As CBD lacks psychotropic effects, products containing only CBD are not legally restricted in many countries. Health authorities such as the US Food and Drug Administration (FDA), the Government of Canada, the European Medicines Agency (EMA), the UK Government and the Australian Government provide ample information on cannabis products on their websites, including legal aspects [39,42,43,44,45].

### 3.2. Cannabis Products for Medical Purposes

Over time, various cannabis products have been developed for medical purposes, and some have meanwhile been approved by national health authorities in a number of countries including Canada, Germany, Israel and the Netherlands [4]. They include pharmaceutical dosage forms containing (synthetic) THC and/or CBD as well as raw herbal materials (known as ‘medicinal cannabis’ or ‘medical cannabis’). Table 1 gives an overview of approved (by national health authorities) medicinal cannabis products, with the corresponding indications and uses. The products encompass pharmaceutical formulations (capsules, oral liquid, oromucosal spray) and medicinal cannabis. The latter category contains raw herbal material (cannabis flos or cannabis inflorescences) and granulates, which should be further processed before administration to a patient. Each medical cannabis variant is genetically characterised, possesses a characteristic terpenoid pattern and is standardised to contain specific amounts of THC/CBD [46,47].

Cannabis-based oil products (commonly known as ‘cannabis oil’) are prepared by further processing the raw material in a specialised preparatory pharmacy or by the pharmaceutical industry. Cannabis oil for medical use (on prescription) may be prepared by supercritical carbon dioxide extraction of cannabis flos or granulates under controlled conditions or by extraction of the plant material with ethanol. Using the first method, an amber-coloured oily fraction containing THC, CBD and terpenes is obtained, separated from the green pigmentation in the plant material. The second method yields a dark brown to black extract containing the cannabinoids. The oil is first heated and then mixed with almond oil or refined peanut oil, after which the product is adjusted to the desired concentration. The chosen variety of cannabis flos or granulates (see Table 1) will determine the qualitative and quantitative composition of the end product and thus its use. In practice, THC, THC/CBD and CBD oils are distinguished. To the latter, if a product contains less than negligible amounts (<0.05%) of THC, limiting narcotics regulations are generally not applicable [47,54]. Recently, it was reported that CBD in an acidic environment may degrade to yield psychotropic products including THC. This finding may be relevant for the storage stability of cannabis oil [32].

A placebo cannabis oil product may be prepared by using the residual plant material after an initial ethanol extraction from the cannabis flos granulate, followed by a second extraction with the same solvent. A placebo containing <0.5% cannabinoids is available for use in clinical studies with cannabis oil standardised on 10% THC and 5% CBD [55]. 

All preparation processes contain a heating step. Cannabinoids are present in the plant as carboxylic acids, which undergo decarboxylation upon heating. THC and CBD are decarboxylated products [56]. When brewing a tea from raw material, the temperature may not be high enough, resulting in incomplete decarboxylation and a concomitant lower biological activity. Interestingly, the acids were shown to be able to block cellular entry of severe acute respiratory syndrome coronavirus-2 (SARS-CoV-2) and are receiving interest in the search for small molecules preventing COVID-19 [57].

A patient may use medicinal cannabis in several ways (see also Section 3.4), by vaporising followed by inhalation, or by brewing a tea. In all cases, it is important that specific directions for the use are provided by the supplier to guarantee an optimal treatment. 

### 3.3. Quality of Cannabis Products

As the target groups for medicinal cannabis include critically ill patients, they should not be exposed to risks associated with poor-quality products. The quality of cannabis products is therefore essential to consider [47,58,59]. Generally speaking, medicinal cannabis is of considerably higher and more constant quality than recreational cannabis purchased from a coffee shop. In addition to the higher quality ensured by rigorous control, medicinal cannabis and products thereof should be characterised qualitatively and quantitatively, and be standardised on a given and fixed cannabinoid content (see Table 1). This is of utmost importance for clinical studies and safety assessment. 

In many countries, cannabis and cannabis products are freely accessible ‘over-the-counter’ from drug stores and coffee shops (within the limits of legislation), while medicinal cannabis and products thereof are dispensed from the government and/or through a (community) pharmacy [60,61]. Health claims made by manufacturers of over-the-counter products cannot be simply accepted by medical professionals, since supporting clinical research is lacking in most cases, dose measurements are determined by consumers, and labelled information about cannabinoid content is often incorrect or even missing [61,62]. 

Hazekamp [60] investigated the quality of street-market-purchased cannabis in the Netherlands and found that ten out of ten obtained samples exceeded the limits for microbiological purity for inhalation products, as set by the European Pharmacopoeia (EP). This is relevant, as most frequently, cannabis for recreational purposes is self-administered by smoking. In contrast, two pharmaceutical-grade cannabis products, obtained from the government of the Netherlands, met the requirements of the EP regarding microbial purity. A more thorough analysis of one of the street-market samples even revealed the presence of the intestinal bacterium *Escherichia coli*, as well as fungi of the genera *Penicillium*, *Cladosporium* and *Aspergillus*. This study also revealed large variations in cannabinoid content among street-market products. In an American study, health insurance claim data from 2016 were used to evaluate the occurrence of fungal infections in cannabis users (53,217) compared to non-users (21,559,558). Cannabis users appeared to be 3.5 times (95% CI 2.6–4.8) more sensitive than non-users. In this study, however, no casualty between cannabis users, the cannabis product and the fungal infection could be established, but precaution with immunocompromised patients is definitely warranted [63]. All those findings underpin the importance of the use of only pharmaceutical-grade medicinal cannabis for critically ill patients, including patients with cancer. 

A recent analysis of 293 cannabis products, mostly CBD oils, on the German market revealed that 10% of these products contained THC above the lowest observed adverse effect level, which was set at 2.5 mg/day [64]. 

Cannabis oil is available from commercial sources via different channels or can even be homemade. Such oils are mostly not controlled in the laboratory and may be of poor quality. They may (almost) lack cannabinoids and contain heavy metals, harmful contaminants and solvents. 

Thus far, no specific monograph on cannabis raw material and/or cannabis products is available in the EP or in the United States Pharmacopeia (USP). For medicinal cannabis, the general requirements described in the monograph on herbal medicinal products in the EP are applicable. Production of medicinal cannabis should take place strictly under Good Manufacturing Product (GMP) conditions [65]. The European Medicines Agency (EMA) recently published a compilation of terms and definitions for cannabis-derived medicinal products [58]. The USP recently published considerations for quality attributes regarding cannabis inflorescences for medical purposes [66]. The EMA is currently developing an herbal monograph on cannabis flos [67].

### 3.4. Administration Routes for Medicinal Cannabis Products

Medicinal cannabis is taken by inhalation, orally or by sublingual administration [68]. Table 2 presents an overview of administration routes for medicinal cannabis products, the recommended dose and the dose frequency, which are subsequently discussed.

Medicinal cannabis can be taken orally in the form of a tea. Tea is brewed by adding 1.0 g of cannabis flos to 1.0 L of boiling water followed by gentle simmering for 15 min [54]. The mixture is strained and the tea can be consumed. Tea can be kept in the fridge for maximally 5 days. Cannabis tea contains relatively low amounts of cannabinoids because of their poor aqueous solubility. The addition of fatty food products such as milk, milk powder, butter or oil to the tea enhances the solubility of cannabinoids and thus their concentration in the tea.

Inhalation is performed by smoking a cigarette containing raw cannabis material mixed with tobacco, or by vaporisation using a vaporiser (‘vaping’). Smoking medicinal cannabis is strongly discouraged because pyrolysis products can damage the lungs and cause inflammation of the throat, nose and lungs. Vaporisation can be applied by non-smokers as well. Vaporisers are medical aids of which various types are on the market. A representative example is the Volcano Medic which is available in many countries. Raw cannabis material is introduced into the vaporiser and undergoes controlled heating. This results in the evaporation of the cannabinoids in the form of an aerosol, without the release of toxic plant combustion substances. The aerosol is collected into a balloon, immediately cooled and subsequently inhaled [70]. Recently, a metered-dose inhalation system was developed for cannabis inhalation, marketed under the brand name SyqeAir. This medical aid works with a cartridge delivering vapour with a precise and adjustable dose of medicinal cannabis. It is recommended to obtain a reduction in the intensity of chronic neuropathic pain, thus for symptom management [71].

Cannabis oil is used by applying droplets sublingually. The lipophilic cannabinoids are rapidly absorbed over the oral mucosa to yield a systemic effect. The oil may be swallowed after a minute. 

## 4. Clinical Aspects of Cannabis Products

### 4.1. Pharmacokinetics of THC and CBD

Comparing the different administration routes and dosage forms, considerable differences in the pharmacokinetics of THC and CBD are seen. In addition, pharmacokinetic data reported in the literature for the same cannabinoid and the same administration route show variations. Most is known about THC, while the pharmacokinetics of CBD seems more variable and unpredictable. The pharmacokinetics of THC and CBD have been extensively reviewed [72,73,74,75]. Table 3 gives an overview of the main pharmacokinetic data, which are subsequently discussed shortly.

When cannabis is taken orally, the bioavailability of THC and CBD is low, due to a substantial first-pass metabolism. The effect of THC is noticed after 30–90 min, and maximal plasma concentrations are reached after several hours. The effect lasts for 4–8 h. After vaporisation (or smoking) of cannabis, THC and CBD are rapidly absorbed via the pulmonary epithelium. Their bioavailability is higher than that after oral intake. Bioavailability depends on the depth of inhalation, the lung capacity, the waiting time between inhalations, the number of inhalations, the time the inhaled breath is held and the heating temperature [69]. Loss is due to the exhaled fraction. Maximal plasma concentrations are reached within 10 min, while psychoactive effects are already noticed within several minutes. After sublingual administration of cannabis oil, the bioavailability of THC and CBD is comparable to that of the oral route. An effect is seen after 30 min and lasts for 4–8 h, with maximal plasma concentrations after several hours [69,72,73,74,75].

All commonly used administration routes have advantages and limitations. The bioavailability of THC and CBD is always low. Absorption via the oral route may be unpredictable, but the effect lasts long. The taste of cannabis tea and cannabis oil may be experienced as unpleasant and may lead to acceptance problems by the patient. However, the acceptance of cannabis oil by patients is generally good. An inhalation device is quite expensive and may be a burden for the user. Inhalation is unsuitable for children. Other possible routes of administration for cannabis products that are being considered include rectal and transdermal, but they require further research. 

THC and CBD are largely bound to plasma proteins, predominantly lipoproteins and to a lesser extent albumin [75]. The apparent volume of distribution is high: for both THC and CBD, it is estimated to be around 30 L/kg [69,72]. THC rapidly divides over well-vascularised organs and eventually in fat tissue where it accumulates, thereby enhancing the elimination half-life [76].

Phase I metabolism of THC and CBD in the liver involves microsomal hydroxylation and oxidation reactions, carried out by cytochrome P450 isoenzymes. The principal isoenzymes involved are CYP3A4, CYP2C9 and CYP2C19 for THC, and CYP3A4 and CYP2C19 for CBD [69,72,77]. This may be followed by phase II glucuronidation by UDP-glucuronosyltransferases (UGTs).

The elimination half-life of THC and CBD varies substantially among cannabis users. Heavy users show the longest elimination half-life, ranging from one to several days. The longer half-life is due to the accumulation of THC and CBD in fat tissue because of their high lipophilicity. They are subsequently slowly released from the fat tissue, e.g., by redistribution or due to weight loss of the user [72].

In the literature, conflicting data are found as to whether CDB may or may not interfere with the bioavailability and/or metabolism of THC. It has been reported that CBD can either attenuate or increase the effects of THC and that a pharmacokinetic interaction may occur through cytochrome P450 inhibition. Findings from a recent randomised clinical trial (RCT) with 18 healthy adults, in which the effects of oral 20 mg THC plus oral 640 mg CBD were compared to 20 mg THC alone and placebo, suggest that high doses of CBD can inhibit the metabolism of THC, resulting in stronger drug effects (greater impairment of cognitive and psychomotor activity, greater increase in heart rate) [78].

### 4.2. Cytostatic Activity of Cannabinoids and Cannabis Products

Since the appearance in 1975 of the first paper describing the cytostatic effects of cannabinoids in mice [79], a substantial number of preclinical studies using in vitro and in vivo models on this topic have been published. Dose-dependent growth inhibition has been reported in various tumour cell systems and in animal tumour models. Insight has been obtained that cannabinoids interfere with signal transduction pathways of the endocannabinoid system involved in tumour progression [18,27]. Cell cycle arrest; induction of apoptosis; inhibition of angiogenesis; stimulation of autophagy; and inhibition of migration, invasion and metastasis have been reported [22,37,80]. Recently, a synergistic action in MCF7 cells was found for CBD combined with docetaxel, paclitaxel, doxorubicin, vinorelbine or 7-ethyl-10-hydroxycamptothecin (SN-38, an active metabolite of camptothecin) [81]. 

Mitoxantrone or vinblastine combined with CBD in canine urothelial cell lines significantly reduced cell viability and increased apoptosis compared to single-drug treatment in a synergistic way, as shown with combination index calculations. The combination of CBD with cisplatin did not reveal a stronger cytostatic action in these cells [82]. Combined therapy with THC and temozolomide in mice with glioma xenografts was more effective in tumour growth reduction than temozolomide alone [83]. Gemcitabine combined with cannabinoids strongly inhibited the growth of pancreatic tumour cell xenografts in nude mice through a reactive oxygen species (ROS) mechanism [84]. The anticancer effect of ionising radiation in an orthotopic murine glioma model was enhanced by CBD and THC [85]. In recent years, it has become clear that interactions of cannabinoids with the immune system may also play a role in their effects on cancer progression. For detailed surveys of these topics, we refer to reviews by [9,12,86].

Well-designed clinical trials are needed to demonstrate whether and to what extent the wealth of available preclinical data can be translated toward positive treatment results in patients with cancer. At present, reliable clinical data towards the cytostatic activity of cannabis products are scarce. 

Two papers reported on presumed positive effects of CBD in patients with cancer, but the studies are small trials, weak in terms of power, and seem merely anecdotical case reports [87,88]. Kenyon et al. [87] analysed data collected over a four-year period of 119 patients with different solid tumours. Of them, 28 were treated with pharmaceutical-grade synthetic CBD in oil only, and the rest received CBD in combination with other therapy. In 92% of all 119 cases, positive clinical responses were seen. Treatment with CBD consisted of three days on and three days off regimen with an average dose of 10 mg twice a day, depending on tumour size. Worth mentioning is the case of a 5-year-old male patient with no available treatment left for his anaplastic ependymoma, showing approximately 60% tumour reduction after treatment with CBD. Another case concerned a 50-year-old patient diagnosed with progressive tanycytic ependymoma grade 2, showing tumour size reduction after treatment with CBD. A remarkable result according to the authors was that, when a switch was made from pharmaceutical-grade CBD in oil to internet-obtained cannabis oil of unknown quality and composition, the disease progressed again.

Dall’Stella et al. [88] reported two cases of adult males with confirmed diagnosis of high-grade gliomas (grades III/IV), who were treated with procarbazine, lomustine and vincristine (PCV) in combination with CBD after subtotal resection of the tumour. In addition, THC was inhaled during the first year of treatment. PCV is known to be highly toxic in haematological and hepatological areas. Yet, the two patients remained without disease progression during the study time and did not show any serious adverse effects during PCV treatment. One patient, however, developed an exacerbated inflammatory response soon after chemoradiation, but significant adverse effects as expected were not seen, which, according to the authors, was probably due to the anti-inflammatory effects of CBD. 

A recent study with more power than the two above-mentioned cases is the ‘Glioblastoma phase 1b trial’ [89] wherein the efficacy and safety of oromucosal nabiximols spray (containing synthetic THC and CBD, see Table 1) in combination with dose-intense temozolomide (DIT) in glioblastoma patients were investigated. Although the sample size was small and therefore results must be considered carefully, the overall survival at 1 year of 83% in the nabiximols-treated group versus 44% in the placebo-treated group was according to the authors considered promising. In addition, adverse events from chemotherapy such as vomiting, dizziness, fatigue, nausea and headache were reduced. During this trial, nabiximols spray was dosed in a personalised dosing manner, therefore limiting adverse events as much as possible and still achieving promising overall survival results. No interaction of any kind was seen during the concomitant use of nabiximols and DIT, indicating the safety of cannabinoid use during regular cancer therapy. The efficacy of cannabinoids will be further investigated in an upcoming phase 2 trial. In this trial, glioblastoma patients will be treated with temozolomide and in combination with Sativex (oromucosal spray containing THC and CBD 1:1, see Table 1) or in combination with placebo [90].

In New Zealand, a phase 2 RCT was carried out with patients diagnosed with high-grade glioma. A single nightly dose of THC-containing medicinal cannabis was safe, had no serious adverse effects and was well tolerated. The cannabis product improved sleep, functional well-being and quality of life. Interestingly, a tumour reduction (based on RANO criteria) was found in 9% (8/88) of the patients 12 weeks after combining standard treatment with THC/CBD oil [91].

In Spain, a phase IB open-label, multicentre, interpatient dose escalation clinical trial to assess the safety profile of a THC/CBD combination with temozolomide and radiotherapy in patients with newly diagnosed glioblastoma is expected to start in the course of 2023 [92].

Currently, a pilot study at the University Medical Center Groningen (the Netherlands) is being conducted, investigating the oncologic potential of cannabis oil, containing both THC (10%) and CBD (5%), in hepatocellular carcinoma patients, in cases where no other treatment is optional (EUDRACT 2018-004505-34) [93].

### 4.3. Symptom Relief by Cannabis Products

Patients with cancer frequently suffer from disease-related symptoms, such as pain, nausea and vomiting, loss of appetite and weight loss, mood swings and sleep disorders. In addition, the treatment with anticancer drugs is associated with adverse effects that may negatively influence a patient’s quality of life [94]. Cannabis products show potential in ameliorating symptoms originating from the disease as well as to counteract certain adverse effects caused by anticancer therapy [13,95]. Encouraging results have been described in the treatment of cancer pain [96]; in reducing chemotherapy-induced nausea and vomiting, anorexia and neuropathic pain [94,97,98]; and in reducing opioid use for pain control [99,100]. Case reports pointing to the potential of cannabinoids to improve a patient’s quality of life during the palliative treatment of cancer have been reviewed [101].

In addition, it has been shown in preclinical studies that concomitant use of cannabis products may have a protective effect against oral and, more generally, gastrointestinal mucositis [102,103,104], one of the most common and serious adverse effects of chemotherapy [105]. Mucositis results in epithelial injury and apoptosis due to the production of reactive oxygen species (ROS) and DNA damage and causes difficulties with eating. As a result, a patient may no longer tolerate the offered therapy, requiring dose reduction or even discontinuation [104].

It was recently reported that 25% of oncological patients with solid malignancies used cannabinoids for medical purposes during active anticancer treatment, and 18% considered doing so. In other words, an increasing number of oncological patients consider cannabis products in order to relieve their symptoms [106]. A cross-sectional survey among 3435 Danish patients identified a higher cannabis use for self-treatment (mostly with cannabis oil) in smokers, in patients who had active cancer treatment and in patients with depression. In this study, cannabis use even correlated with a lower quality of life (EORTC C30 Sum score mean diff. = −7.61, 95% CI = [−9.69; −5.53]). However, 77% of the cannabis users experienced at least one positive effect (management of pain, less nausea, better sleep) [107]. 

Individual observational studies demonstrated significant improvement in patient-reported outcome measures. From a prospective noncomparative registry of 2991 cancer and non-cancer patients who initiated medical cannabis, a significant improvement in patient-reported symptoms appeared as measured by questionnaires, including pain, nausea and well-being [108]. In a double-blind randomised placebo-controlled trial, it was demonstrated that naboximols improved the average pain numerical rating scale (NRS) score in advanced cancer patients with chronic uncontrolled pain (10.7% improvement in naboximols patients versus 4.5% improvement in placebo patients, *p* = 0.085, intention to treat analysis). In per-protocol analyses, the differences were even higher (15.5% improvement in naboximols patients versus 6.3% improvement in placebo patients, *p* = 0.038) [109].

In an Israeli study, a prospective analysis of the safety and efficacy of medicinal cannabis in an unselected population of 2970 patients with cancer, it was shown that cannabis as a palliative treatment was well-tolerated, effective and safe, and aided patients with malignancy-related symptoms [110]. 

Nabilone, a synthetic THC analogue and FDA-approved for refractory or breakthrough (chemotherapy-induced) nausea and vomiting, was shown to be effective in increasing carbohydrate intake in a randomised, double-blind, placebo-controlled clinical trial in lung cancer patients focusing on appetite, nutritional status and quality of life [111]. In this clinical trial, health-related quality of life was assessed by questionnaires (EORTC-QLQ-C30 and QLQ-LC13) and shown to improve significantly when focusing on role functioning, emotional functioning, social functioning, pain and insomnia. Moreover, adding oral THC/CBD to standard antiemetics was associated with less nausea and vomiting in a randomised, placebo-controlled phase II crossover trial in patients with refractory chemotherapy-induced nausea and vomiting [112]. In this trial, patients in the THC/CBD group showed a significantly higher proportion (25%) with complete response (no vomiting and no used rescue medications during 0–120 h from chemotherapy) compared to the placebo group (14%, *p* = 0.04). In addition, the percentage of patients without vomiting, without the use of rescue medications and without significant nausea was significantly higher in the THC/CBD group. 

Nevertheless, several recent meta-analyses that included both RCTs and non-randomised studies demonstrated no effect of cannabis products on appetite increase or improvement of quality of life in cachectic cancer patients [113,114,115]. Additionally, from a systematic review and meta-analysis including five randomised trials (all enrolled patients suffered from chronic cancer pain) and twelve observational studies focusing on opioid-sparing effects of cannabis products in patients with chronic pain, it was concluded that adding cannabis had little or no effect on pain relief or sleep disturbance [116]. The authors even observed a likely increase in nausea (relative risk (RR) 1.43; 95% CI 1.04 to 1.96; risk difference (RD) 4%, 95% CI 0% to 7%) and vomiting (RR 1.5; 95% CI 1.01 to 2.24; RD 3%; 95% CI 0% to 6%) upon the use of cannabis products. 

This was corroborated by another recent systematic review with meta-analysis and Trial Sequential Analysis, comprising 65 randomised placebo-controlled clinical trials in which a total of 7017 participants were enrolled [117]. This analysis revealed no effect of cannabinoids on reducing acute pain or cancer pain, nor improvement of the quality of life. Although cannabinoids reduced chronic pain and improved the quality of sleep, the effect sizes were disputable. Additionally, it appeared from this analysis that cannabinoids may increase the occurrence of non-serious adverse events induced by anticancer drugs, but not serious adverse events. This may be ascribed to the difficulty of precise cannabinoid dosing in human patients and shows the need for individualised cannabinoid dosing using upward dose titration in order to avoid overdosing [118,119]. 

In a randomised, placebo-controlled, dose-escalating, double-blind phase IIB trial, the effect of CBD oil was studied in 144 patients with advanced cancer and receiving palliative care [120]. No detectable effect of CBD oil was found on the quality of life, depression or anxiety.

In conclusion, although cannabinoids may be effective in symptom management for selected oncological patients and for selected complaints, no significant overall clinical effect of cannabinoids in symptom relief was observed across the various available studies. Additionally, the use of cannabis products in combination with anticancer drugs may result in more non-serious adverse events, possibly due to overdosing. The claimed advantages of applying cannabis products for symptom management and relief in patients with cancer remain controversial. 

### 4.4. Possible Interactions of Cannabis Products with Anticancer Drugs

When using medicinal cannabis products, there is a risk that interactions occur with anticancer drugs, influencing the therapeutic outcome. Depending on the nature of the interaction and the type of anticancer drug, the cytostatic effect of anticancer drugs may in theory be either mitigated or enhanced. 

In the literature, the focus has been primarily on pharmacokinetic interactions between cannabinoids, cannabis products and anticancer drugs, affecting their disposition in the body. Our current knowledge mainly comes from preclinical research, in vitro models using cell systems or recombinant enzymes, while concentrations of the cannabinoids used (THC and CBD) often exceeded those expected in serum after cannabis use. Hence, the translation of the in vitro data to the clinical situation is difficult, but an understanding of the underlying mechanism will at least enable us to predict what might happen if we use cannabis products with concomitant anticancer therapy and thus to better perform a risk–benefit assessment of such combined treatment. Clinical evidence for drug–cannabis interactions, however, is largely lacking so far [77,121,122].

In principle, cannabis products may modulate the pharmacokinetic profile of anticancer drugs by interfering with drug transporters and with metabolic enzymes. A very important, yet in many studies neglected, aspect here is the qualitative and quantitative composition of the cannabis products used, the route of administration and the exposure (dose, frequency and duration of treatment). 

Pharmacodynamic interactions may theoretically occur via competitive or allosteric inhibition of cannabinoid receptors, via interference with the regulation of their gene expression and via interference with transduction pathways connected to the endocannabinoid system. 

#### 4.4.1. Drug Transporter Proteins

The crossing of a number of anticancer drugs over biological membranes is mediated by transmembrane proteins of the ATP binding cassette (ABC) family. Important representatives are P-glycoprotein (P-gp), breast cancer resistance protein (BCRP) and multidrug resistance proteins (MRPs), which function as efflux pumps for a broad range of substrates [121]. Efflux pumps are found in the intestine, liver, kidneys, brain and many tumours. Overexpression in cancer cells contributes to multidrug resistance [123,124]. Altering the efficacy of such transporters may alter the cellular concentration of drug compounds that serve as substrates for them [125].

It has been shown in in vitro studies that cannabinoids are able to interact with P-gp. A short exposure (1 h) of multidrug-resistant cells to THC and CBD resulted in a higher expression of P-gp, while a longer exposure (72 h) decreased expression. This may indicate that longer exposure to cannabinoids may increase the cytotoxicity of anticancer agents that serve as P-gp substrates, such as doxorubicin, methotrexate, docetaxel, vinblastine and many tyrosine kinase inhibitors [121,123,126].

In another in vitro study, elevated cellular concentrations of mitoxantrone and topotecan were measured due to an inhibitory effect of THC and CBD on the BCRP transporter [127]. Furthermore, inhibition of P-gp-dependent efflux has been reported in trophoblast-like cell lines after a 1 h exposure to CBD, along with upregulation of BCRP levels after 72 h of exposure to CBD [128]. 

In vitro increased cellular accumulation of several substrates for MRP transporters, including vincristine, has been reported to occur after exposure to THC and CBD [129]. It has furthermore been shown that CB1 receptor antagonists are able to stimulate the transport activity of MRPs [130].

As various anticancer drugs are substrates for efflux proteins, the ability of cannabinoids to inhibit them may be clinically relevant and used to tailor therapy. Further research into this direction is warranted. Possibly, combining cannabis products with anticancer drugs that undergo a high efflux from tumour cells has a therapeutic advantage in the form of higher cytostatic activity and/or a lower possible dose with concomitant reduction of adverse effects. Furthermore, the absorption of orally administered anticancer drugs may be improved by concomitant treatment with cannabis products acting as inhibitors of intestinal efflux pumps. 

Cannabinoids show a high degree of plasma protein binding. Hence, an interaction may occur on this level with anticancer drugs that are highly bound to plasma proteins, including platinum drugs (e.g., cisplatin) and 5-fluorouracil [131,132,133]. On the one side, the free fraction of an anticancer drug may increase due to replacement, resulting in higher serum levels of the anticancer drug and an enhanced risk of adverse effects. However, given the low plasma concentrations reached for THC and CBD, this effect is expected to be limited. On the other side, the opposite may occur: THC and CBD may be displaced from protein binding by anticancer drugs.

#### 4.4.2. Metabolising Enzymes

THC and CBD are substrates for various CYP450 isoenzymes in the liver, thereby undergoing oxidation reactions (phase I metabolism). In in vitro studies with recombinant enzymes and human kidney and liver microsomes, these cannabinoids displayed inhibition of CYP2A6, CYP2B6, CYP2C8, CYP2CP, CYP2D6, CYP3A4 and CYP3A5. While the inhibitory concentrations of THC and CBD against most isoenzymes were much higher than the serum concentrations of cannabis users, the concentrations of cannabinoid-induced inhibition of CYP3A4 and CYP3A5 were in a comparable range [121,134,135,136,137,138,139]. Furthermore, in vitro inhibition of UGTs (conjugation reactions; phase II metabolism) by cannabinoids has been reported [140].

Many anticancer agents undergo metabolism via CYP450 isoenzymes (especially CYP3A4) and a number of these metabolites are conjugated (mainly by UGTs) prior to excretion [126]. Inhibition of CYP450 isoenzymes and/or UGTs, induced by cannabis products, may increase the serum concentrations of anticancer agents that are substrates for these enzymes. On the one hand, a dose reduction of the anticancer drug may be required to avoid excessive toxicity. On the other hand, such interaction may theoretically be used to lower the dose of the anticancer drug.

Anticancer drugs that are converted into one or more inactive metabolites by CYP450 isoenzymes include the anti-androgen abiraterone, the aromatase inhibitor anastrozole, the taxanes docetaxel and paclitaxel, the vinca-alkaloid vincristine, the intercalant doxorubicin and the topoisomerase inhibitor irinotecan [141,142]. UGTs play a role in the elimination of anastrozole, doxorubicin, epirubicin and irinotecan [69].

An opposite effect may be encountered if the anticancer drug administered is a prodrug that is converted into an active metabolite or active metabolites by CYP450 isoenzymes. Anticancer drugs in this category include the alkylating agent cyclophosphamide, the anti-oestrogen tamoxifen, and the tyrosine kinase inhibitors sorafenib and regorafenib [69,143]. Inhibition of CYP450 isoenzymes may reduce the serum concentration of active metabolites, thereby reducing the cytostatic effect of these drugs.

The vinca-alkaloid vinblastine and the topoisomerase II inhibitor etoposide display cytostatic effects themselves and are converted into biologically active metabolites by cytochrome P450 isoenzymes [69]. Inhibition of cytochrome P450 isoenzymes by cannabis products may influence the ratio of the parent drug and the active metabolites, all displaying cytostatic effects.

For anticancer drugs that are not substrates for cytochrome P450 isoenzymes and/or UGTs, no such pharmacokinetic interaction is expected in combination with cannabis products. In this category fall the folic acid antagonist methotrexate, the pyrimidine antagonists fluorouracil and gemcitabine, the CDK46 inhibitor abemaciclib, the platins cisplatin and carboplatin, and monoclonal antibodies. The elimination of monoclonal antibodies occurs mainly through intracellular catabolism to amino acids (by proteases) after uptake by pinocytosis or by receptor-mediated endocytosis [144].

Thus, in theory, pharmacokinetic interactions between cannabinoids and anticancer drugs may occur, but the clinical impact (if any) is yet unclear. For instance, in a phase IB randomised, placebo-controlled trial of nabiximols with temozolomide in patients with recurrent glioblastoma, no effects of nabiximols were found on the pharmacokinetics of temozolomide [89].

#### 4.4.3. Immunotherapy

Next to possible pharmacokinetic interactions between cannabinoids and metabolising enzymes or transporter proteins, there is a potential pharmacodynamic interaction with immune checkpoint inhibitors (ICIs). These immunomodulatory anticancer drugs, which include the monoclonal antibodies pembrolizumab, ipilimumab, nivolumab and atezolizumab, interfere with checkpoint proteins: cytotoxic T-lymphocyte-associated protein 4 (CTLA4), programmed cell death protein 1 (PD-1) and programmed death-ligand 1 (PD-L1) [145]. Checkpoint proteins signal the immune system, resulting in the activation or deactivation of the immune response, in order to attack harmful cells and protect healthy cells. Tumour cells may inhibit this form of communication with the immune system, resulting in the inhibition of a T-cell response against the tumour cells. ICIs can counteract this inhibition, resulting in T-cell activation and amplification of an immune response [146]. 

In a retrospective observational study (2015–2016), including 140 hospitalised patients with advanced melanoma, non-small-cell lung carcinoma or renal clear cell carcinoma, 89 patients were treated with nivolumab and 51 with nivolumab plus cannabis products. These products were supplied by six different companies and their composition was unknown. In patients treated with the combination, the response rate (RR) to immunotherapy was decreased compared to that of the group receiving nivolumab alone (37.5% RR in nivolumab alone compared with 15.9% in the nivolumab–cannabis group (*p* = 0.016, odds ratio = 3.13, 95% confidence interval 1.24–8.1). No significant difference in effect was found regarding progression-free survival (PFS) and overall survival (OS) between the two groups [147].

In a prospective observational study (2016–2018), the effect of ICIs on concomitant cannabis use among patients with advanced cancers (metastatic malignancies, stage IV disease) was investigated [145]. In total, 102 patients initiating checkpoint inhibitor treatment were included. Of this group, 34 patients used cannabis while 68 did not. Miscellaneous and not further characterised cannabis products were used. All patients used less than 40 g of cannabis products per month. A correlation was found between cannabis use and a decrease in time to tumour progression and overall survival. The median time to tumour progression (TTP) was 3.4 months (95% confidence interval 1.8–6.0) for cannabis users and 13.1 months (95% confidence interval 6.0-NA) for non-users. The overall clinical outcomes were estimated in terms of complete response (CR), partial response (PR) and stable disease (SD) according to the Response Evaluation Criteria in Solid Tumors RECIST 1.1 criteria. Cannabis users showed a significantly lower percentage of clinical benefit (CR + PR + SD) outcomes: 39% for users and 59% for non-users (*p* = 0.035).

The observed worsened clinical outcome due to this interaction may be explained by the immunomodulatory effects of cannabinoids. It was also found that the use of cannabis reduced therapy-related immune-related adverse events. Cannabinoids, and particularly THC, have the potential to suppress CD8 T cells and cytotoxic T-lymphocyte activity. In addition, the proliferation of lymphocytes and maturation to mature cytotoxic T lymphocytes may be inhibited by THC [145]. 

Although the above-mentioned studies were observational and non-randomised, patient selection may have occurred, it may be concluded that caution must be taken with cannabis users to be treated with immunotherapy. Cannabis may negatively influence the therapeutic outcome due to its immunomodulatory properties [145,147]. It may, however, be possible that patients using cannabis were precisely those with a more complex symptomatology and worse overall health status/prognosis.

### 4.5. Possible Interactions of Cannabis Products with Other Medication Used by Patients with Cancer

Patients with cancer often use a plethora of medicines: to relieve pain caused by the disease; to counteract chemotherapy-induced adverse effects such as nausea, vomiting and anorexia, to treat obstipation induced by opioids or diarrhoea caused by anticancer medication; and to treat sleeping problems and mood disorders. An important therapeutic goal is the improvement of the quality of life, especially in the palliative phase [148]. 

Interactions with drug-metabolising enzymes, as described above for anticancer drugs, may occur between cannabinoids and supporting medication as well and influence the therapeutic outcome and possibly the choice of a representative from a given therapeutic class. Based on the existing knowledge about the metabolism of drugs frequently used for patients with cancer and the fact that cannabinoids are inhibitors of many cytochrome P450 and UGT isoenzymes, we may predict for which drugs extra awareness is warranted and of which drugs the pharmacokinetic profile is unlikely to be influenced by cannabis products. Yet, the clinical relevance is in most cases unclear and remains to be established.

Patients with cancer frequently use pain medication, including paracetamol, non-steroidal anti-inflammatory drugs (NSAIDs) and opioids. Paracetamol, ibuprofen, diclofenac, naproxen, fentanyl, oxycodone and tramadol all undergo phase I metabolism by cytochrome P450 isoenzymes. Morphine is a substrate for UGTs, while UGTs are also involved in the phase II metabolism of diclofenac, naproxen and tramadol. Recently, ketamine has been receiving attention in the treatment of chronic cancer pain. It is largely converted by cytochrome P450 enzymes in the liver [149]. As a result of the inhibition of the metabolising enzymes by cannabis products, dose reduction of analgesics without loss of analgesic activity might be possible.

Corticosteroids such as dexamethasone, prednisone, prednisolone and triamcinolone are used to counteract nausea and vomiting. All are substrates for CYP3A4, which is inhibited by THC and CBD. In addition, dexamethasone is a substrate for P-gp, which is also inhibited by cannabinoids. Other antiemetics used by patients with cancer are dopamine receptor antagonists (e.g., metoclopramide), 5-HT3 receptor antagonists (e.g., ondansetron) and NK-1 receptor antagonists (e.g., aprepitant) [150]. Metoclopramide, ondansetron and aprepitant are substrates for cytochrome P450 enzymes. Metoclopramide and aprepitant are also substrates for UGTs. Haloperidol is used for the same purpose and additionally has neuroleptic properties that may benefit a patient with cancer. Haloperidol is largely converted by CYP3A4 in the liver. 

As sleeping aids, benzodiazepines are widely used for patients with cancer [151]. Zolpidem and zopiclone are metabolised by cytochrome P450 isoenzymes; temazepam and lorazepam are metabolised by UGT isoenzymes. Midazolam plays an important role in palliative sedation and is metabolised via CYP3A4 and UGTs. 

Loperamide, used to treat chemotherapy-induced diarrhoea, is a substrate for P-gp and metabolised by cytochrome P450 isoenzymes. Patients with cancer may suffer from obstipation as well, which is often opioid-induced. They are given laxatives such as macrogol and lactulose. As these laxatives are not absorbed, there will be no interaction on the level of drug metabolism [151,152].

Antihistamines, such as cetirizine, loratadine and fexofenadine (H1 receptor antagonists), are used to prevent allergic reactions that may be elicited by anticancer drugs. They currently receive attention because they may enhance the outcome of immunotherapy [104]. Loratadine is largely metabolised by cytochrome P450 isoenzymes, and cetirizine is limitedly metabolised by cytochrome P450 isoenzymes. Fexofenadine hardly undergoes metabolism prior to excretion.

Finally, specific co-medication may be added to the treatment of certain cancer types. Some of these drugs are substrates for metabolising enzymes and theoretically sensitive to the concomitant use of cannabis products, while others are not. Cyproterone acetate, with anti-androgenic activity, is used in prostate cancer and metabolised by cytochrome P450 isoenzymes. Bisphosphonates, such as clodronic acid, ibandronic acid, pamidronic acid and zoledronic acid, are used to prevent osteoporosis and loss of bone density in (metastasised) bone cancer. Bisphosphonates are not metabolised. The monoclonal antibody denosumab is used for the same purpose. Cinacalcet, used in parathyroid carcinoma and hyperparathyroidism, is a substrate for cytochrome P450 enzymes. Hyperuricemia due to chemotherapy can be managed by the xanthine oxidase inhibitors allopurinol or febuxostat. Allopurinol does not interact or hardly interacts with cytochrome P450 isoenzymes, but febuxostat is largely metabolised by cytochrome P450 well as with UGT isoenzymes [69]. Filgrastim and pegfilgrastim are used to treat neutropenia induced by anticancer drugs. These protein drugs are not metabolised via cytochrome P450 or UGT isoenzymes.

In conclusion, the metabolism of many drugs used in patients with cancer other than anticancer drugs is likely to be inhibited to some extent by the concomitant use of cannabis products. This may in theory enhance the desired effect as well as adverse reactions, and dose reduction may be appropriate in certain cases. However, clinical evidence is not unequivocal on this point. For example, pharmacokinetic investigations in patients with chronic pain showed no significant change in the area under the plasma concentration versus time curve for morphine or oxycodone when combined with vaporised cannabis [153]. In addition, other clinical studies with cannabis and painkillers or antiemetics revealed no pharmacokinetic interactions [154]. On the other hand, a pharmacokinetic interaction has been reported between CBD and clobazam in children with refractory epilepsy (elevated clobazam plasma levels) [155].

### 4.6. Adverse Effects and Safety of Cannabis Products

Cannabis products are usually well tolerated, and the adverse effects are limited [156]. A prospective study among patients with non-cancer pain showed a reasonable safety profile when cannabis was used for a period of one year [157]. In addition, in the limitedly available clinical studies with patients with cancer, cannabis products generally seem to be tolerated well [89,158]. In a phase I randomised, double-blind, placebo-controlled trial in healthy volunteers, a single dose of up to 6000 mg and multiple doses of 1500 mg twice daily were well-tolerated. All adverse events reported were of mild or moderate severity and included diarrhoea, nausea, headache and somnolence [159]. In a large retrospective Canadian study with 9766 adults using cannabis products (mean age 73.2 years), dry mouth, drowsiness and dizziness were reported. CBD-containing cannabis oils were the most frequently used products [160]. A large Israeli prospective study with approximately 10,000 patients using medicinal cannabis showed only a low incidence of serious adverse effects [161]. From a recent study with 46 healthy infrequent cannabis users, no evidence was reported that CBD reduced (or protected against) the adverse effects of THC on cognition and mental health [162]. In another recent study, a randomised, double-blind, placebo-controlled, crossover experiment in adults and adolescents in which the acute effects of vaporised cannabis flower preparations were compared (THC 8 mg/75 kg person versus THC + CBD 8 mg + 24 mg/75 kg person versus placebo), no evidence was found that CBD modulated the toxic effects of THC [163]. Possible addictive effects of cannabis products, especially when containing THC, are less or not relevant in patients with (advanced) cancer.

Pharmacovigilance of cannabis products is still limited, as cannabis use has been and is largely recreational. As legalisation of cannabis products becomes clearer and more nuanced in many countries, the monitoring, detection, assessment and management of adverse reactions associated with medicinal cannabis will develop further [164].

## 5. Conclusions and Future Directions

Over the years, a massive number of studies have been conducted on the topic of ‘cannabis’ in the broadest sense. A plethora of preclinical data is currently available in the literature, and there is a clear interest in the potential of cannabis products in oncology, especially from the side of the patients. The translation of preclinical data to clinical applications of cannabinoids and medicinal cannabis however remains challenging. A key question is whether there is a place for cannabis products in healthcare and, more specifically, in oncology. May cannabis products offer added value and, if so, what is their place in the treatment options that we have for patients with cancer? 

First, we may consider the use of cannabis products for the symptomatic amelioration of disease-related complaints and adverse effects induced by anticancer drugs. Second, we must consider the clinical effect of possible and complex interactions of cannabis products with anticancer drugs and other medicines. Third, we must be aware of unexpected outcomes and thus carefully monitor patients who are given a combination of cannabis products (or who appear to be cannabis users).

As the endocannabinoid system has been shown to play a role in tumour development and growth, it has been suggested that cannabis products may be used because of their cytostatic effect against specific tumours in which cannabinoid receptors are highly expressed [10,18,27]. Application of cannabis products for this purpose should only be performed on the basis of tumour characterisation and profiling. Nevertheless, the application of cannabis products alone to treat cancers has not shown convincing results so far. 

Future clinical research is absolutely warranted to scientifically underpin positive preclinical and/or anecdotal data obtained with cannabis products in patients with cancer. Pilot studies investigating anticancer effects should be undertaken to demonstrate possible antitumour efficiency, which should most reasonably be performed in patients who are in a palliative setting and for whom we have no other treatment opportunities. Such pilot studies may form a basis for larger trials. Based on the outcome of such studies, the potential and limitations of cannabis products for the clinic can be more objectively determined. In glioma patients, the results obtained with cannabis products seem hopeful, but larger randomised clinical trials should be undertaken to demonstrate efficiency.

Based on their possible pharmacokinetic interactions with efflux transporters and drug-metabolising enzymes (inhibition), cannabis products may enable dose reduction of anticancer drugs that are substrates for these proteins. If this leads to a reduction of adverse reactions, cannabis products may improve the patient’s quality of life. On the other hand, such a combination may result in a stronger effect than expected and more pronounced anticancer drug-related adverse effects. Although interactions at this level are likely to occur based on insight into mechanistic backgrounds, the clinical impact and relevance are difficult to estimate.

As supporting medication in oncology patients for symptom management, to achieve better pain control, to counteract chemotherapy-induced nausea and vomiting, and as a sleeping aid or a mood improver, cannabis products may be in place. Although the scientific support for these applications is weak and ambiguous, opportunities for using cannabis products in supportive care for cancer are acknowledged [101]. The subjective experience of patients with cancer may be taken into account for the clinician’s decision regarding the prescription of medicinal cannabis products.

Treatment of cannabis users with immunotherapy (immune checkpoint inhibitors) or combining immunotherapy with cannabis products, for instance, to reduce to treat cancer-related symptoms or adverse effects of treatment, should be avoided (or at least performed with caution) because of a possibly hazardous interaction with the immune system based on the immunosuppressive action of cannabinoids, especially CBD. 

There are different administration routes with corresponding dosage forms available for cannabis products and medicinal cannabis. The sublingual route seems the most preferred route, over inhalation (smoking or vaporising) and tea to drink. Droplets or a spray can be easily self-applied. The taste may be a hurdle for some users, however. Cannabinoids are rapidly absorbed over the oral mucosa, and biopharmaceutics are well predictable. For oromucosal administration, products with synthetic THC and/or CBD are available. Cannabis oil, of which the composition is determined by the raw material from which the oil is prepared, is available for sublingual administration. Current research is directed towards the development of advanced dosage forms with cannabinoids, in order to improve bioavailability [165].

To choose between either a product with (mainly) THC, (mainly) CBD or a mixture of both, the effects of THC on the central nervous system (‘high’ feeling) can be considered. The patient may experience this as unpleasant, but it may also improve the quality of life. The choice of a dosage form with either pure, synthetic cannabinoids or cannabis oil can be based on the knowledge that other constituents of the oil (terpenoids, flavonoids) may add to the therapeutic effect. The use of cannabis products fits well in the current developments to achieve personalised medication.

Finally, the quality of medicinal cannabis products is a pivotal aspect for clinical trials and clinical use and seems often underestimated. In many cases, the composition of the cannabis products used is unknown. A product should be fully characterised and standardised in terms of the qualitative and quantitative content of cannabinoids. Full control of the source material, selected plants with a known and reproducible cannabinoid spectrum, and the production process (GMP) is required. The final product should undergo rigorous controls before it is administered to a patient. A product should be reproducible, allowing clinical and toxicological evaluation including comparisons. 

Regretfully, the information that can be found online about cannabis products is not always correct and reliable for the indicated therapeutic areas and may be scientifically weak [166]. Label accuracy of unregulated CBD products appeared to be insufficient [167]. The choice of a reliable provider is therefore essential. The pharmacist, as an indispensable link in the healthcare chain, should be able to advise.

It must be made clear to healthcare providers that the term ‘medicinal cannabis’ is void unless the nature of the product and its reliability are known and secured. The scepticism among many clinicians regarding the use of cannabis products can only be taken away if we have thorough knowledge about the product, its composition and its properties. With this article, we aim to guide clinicians in making choices regarding the responsible application of cannabis products in a setting where clinical data are (almost) lacking, while the demand of patients with cancer to use cannabis products is increasing. 

## Figures and Tables

**Figure 1 cancers-15-02119-f001:**
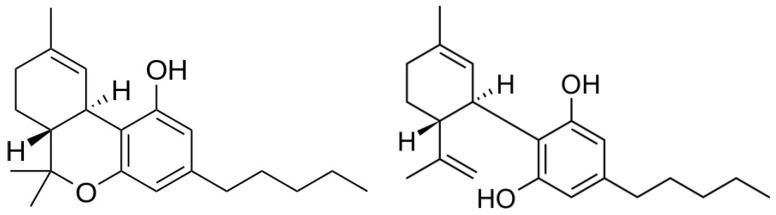
Chemical structures of THC (**left**) and CBD (**right**).

**Table 1 cancers-15-02119-t001:** Overview of approved (by national health authorities worldwide) cannabis products with information about the content of THC and CBD and the corresponding indications and uses.

Cannabis Product	Concentration THC	Concentration CBD	Indications and Uses	Formulation	Reference
Bedrocan	Standardised on 22.0%	< 1.0%	Symptoms include poor appetite, emaciation and vomiting, anorexia, cachexia and emesis. In conditions such as Tourette’s syndrome and therapy-resistant glaucoma.	Cannabis flos	[46,48]
Bedrobinol	Standardised on 13.5%	< 1.0%	As for Bedrocan	Cannabis flos	[46,48]
Bediol	Standardised on 6.3%	Standardised on 8.0%	Inexperienced (with cannabis) patient. Pain, with and without spasm, in patients with MS. Other pathologies with spasms and abnormal muscle activity.	Granulate	[46,48]
Bedica	Standardised on 14.0%	< 1.0%	Patient suffering from restlessness, insomnia or spasms.	Granulate	[46,48]
Bedrolite	< 1.0 %	Standardised on 7.5%	Patients with epilepsy and epilepsy syndromes.	Granulate	[46,48]
Sativex (Nabiximols)	27 mg/mL	25 mg/mL	Spasticity due to multiple sclerosis, in patients who have failed to respond adequately to conventional treatments.	Oromucosal spray	[49]
Epidyolex	-	100 mg/mL	Lennox-Gastaut syndrome and Dravet syndrome as adjuvant therapy. Orphan drug.	Oral liquid	[50]
Cesamet (Nabilone)	Synthetic THC analogue; 1 mg	-	Chemotherapy-induced nausea and vomiting.	Capsules	[51]
Syndros (Dronabinol)	Synthetic THC; 5 mg/mL	-	Anorexia associated with weight loss in patients with AIDS. Nausea and vomiting associated with cancer chemotherapy in patients who have failed to respond adequately to conventional antiemetic treatments.	Oral liquid	[52]
Marinol (Dronabinol)	Synthetic THC; 2.5/5/10 mg	-	Anorexia associated with weight loss in patients with AIDS. Nausea and vomiting associated with cancer chemotherapy in patients who have failed to respond adequately to conventional antiemetic treatments.	Capsules	[53]

**Table 2 cancers-15-02119-t002:** Overview of the administration routes for medicinal cannabis products, the recommended dose and dose frequency [68,69].

Administration Route	Product	Dose (adults) ^1^	Dose frequency ^1^	Remarks
Oral	Tea, prepared from cannabis flos ^2^	200 mL hot or cold tea prepared from 1 g raw material per L	1x daily in the evening	In case of insufficient effect after 1–2 weeks, increase the dose to 200 mL 2x daily (morning and evening)
Pulmonary	Cannabis flos ^2^	To be determined individually	1–2 times daily several inhalations; wait 5–15 min between inhalations	Vaporiser is filled with 100–200 mg raw material
Sublingual	Cannabis oil, prepared from cannabis flos or granulates ^2^	0.05 mL or 1–2 droplets ^3^	2–3 times daily ^3^	For oil containing 20 mg/mL THC and 20 or 13 mg/mL CBD; in case of insufficient effect increase the dose to max 0.25 mL or 10 droplets 3x daily

^1^ For indications see Table 1. Doses and dose frequencies presented count for all indications except for epilepsy. ^2^ See also Table 1. ^3^ To treat epilepsy: Adults 0.1 mg/kg body weight, children 0.25 mg/kg body weight; 2x daily.

**Table 3 cancers-15-02119-t003:** Main pharmacokinetic data of both cannabinoids as found in the literature, in relation to the administration route [69,72,73,74,75].

	THC	CBD
Absorption	Oral T_max_ = 1–5 h F = < 10% Inhalation T_max_ = 3–10 min F = 10–35% Oromucosal/sublingual T_max_ = 1–2 min F = 2–20%	Oral T_max_ = 1–6 h F = circa 6% Inhalation T_max_ = 3–10 min F = circa 30% Oromucosal/sublingual T_max_ = 1.5–5 h F = circa 30%
Distribution	Plasma-protein binding 95–99% V_d_ = 30 L/kg	Plasma-protein binding 99% V_d_ = 30 L/kg
Metabolism	Phase I Oxidation THC → 11-OH-THC→ 11-COOH-THC CYP2C9, CYP2C19, CYP3A4 Phase II Glucuronidation 11-OH-THC by UGT1A9, UTG1A10 11-COOH-THC by UGT1A1, UGT1A3	Phase I Oxidation CBD → 7-OH-CBD → 7-COOH-CBD CYP2C19, CYP3A4 (primarily) and CYP1A1, CYP1A2, CYP2C9, CYP2D6 (to a lesser extent) Phase II Glucuronidation 7-OH-CBD, 7-COOH-CBD by UGT1A7, UGT1A9, UGT2B7
Elimination/ Excretion	t_1/2_ = 1–30 h Mainly as metabolites, in faeces and in urine	t_1/2_ = 1–5 d Mainly as metabolites, in faeces and in urine

Abbreviations: T_max_ = time at which maximal plasma concentrations are reached; F = bioavailability; V_d_ = apparent volume of distribution; t_1/2_ = elimination half-life.

## Data Availability

Not applicable.

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
