# Peer review of "Potential, Limitations and Risks of Cannabis-Derived Products in Cancer Treatment"

_cancers, 2023, doi:10.3390/cancers15072119_

Round 1
Reviewer 1 Report
Cannabinoid research has strongly advanced during the last decades owing to the structural and functional characterization of the endocannabinoid system. Unfortunately, this has not been paralleled in general by consistent human studies, so the potential therapeutic applications of cannabinoids are still a matter of discussion. As marijuana is a natural source of cannabinoids, this debate overcomes the scientific limits and reaches the social arena. Here, Woerdenbag et al. deal with one of the hot topics of current clinical cannabinoid research, namely the possible applications of these compounds in cancer care. Although there are many recent reviews published on the topic, this one includes a fair analysis of the characteristics, quality, and pharmacology of cannabis products, which, to the best of my knowledge, had not been conducted before in the context of the oncology patient. The subject of the paper is therefore appropriate, and so is the idea of discussing the possible pros and cons of cannabinoid-based therapies. There are nonetheless several points that, in my opinion, could help to improve the paper.
1. Lines 102-127 (and rest of the manuscript). To avoid confusion, the authors should clarify the definition and use of the term “cannabinoid”. Is a cannabinoid simply a compound with a THC-like structure, or is it a compound with THC-like pharmacodynamics, i.e., acting on cannabinoid receptors (as currently accepted by the IUPHAR)? Many compounds with a THC-like structure are present in cannabis (including CBD, most of whose pharmacological effects are conceivably independent of its very weak direct binding to CB1R/CB2R) or have been synthesized in the lab (e.g., ajulemic acid), but they do not have a THC-like mechanism of action –in fact, the pharmacological properties of most phytocannabinoids have still to be defined.
2. Lines 128-138. There is still no satisfactory proof for the existence of such an “entourage effect” in humans, for example a solid RCT comparing a full-spectrum cannabis product with the equivalent pure-THC product. This should be clarified.
3. Line 170, Table 1. Nabilone is not “synthetic THC” but a synthetic THC analogue.
4. Line 251, Table 2, and line 279. The term “Oromucosal” should be applied solely to sprays as Sativex. Cannabis oils should be included under “Oral” or, if preferred, under “Sublingual”.
5. Lines 296-327. The are conflicting data in the literature as to whether CBD interferes or not with THC metabolism/bioavailability. The authors might include a short discussion on this relevant issue.
6. Lines 336-343. Other related preclinical studies on the combination of cannabinoids with antineoplastic drugs (e.g., PMID 21220494, PMID: 21525939) as well as with RTx (PMID: 25398831) could also be cited.
7. Line 393. Might it also be worth mentioning PMID 34094937? And clinicalTrials.gov #NCT03529448?
8. Lines 422-431. PMID 29482741 could also be acknowledged.
9. Lines 578 and 694 (and/or “Conclusions and Future Directions” section). A short conclusion/opinion on these two subsections would be valuable for the reader. After all, do the authors believe that those theoretical/potential drug-drug interactions constitute an important hazard for a real-life cancer patient taking cannabinoids? For example, no overt PK interaction of cannabinoids with temozolomide (PMID 33623076) or opiates (PMID 22048225) was found, and, to the best of my knowledge, no remarkable PK interaction (as clear as, for example, that of CBD with clobazam in paediatric epileptic syndromes) has come out in any of the countless clinical studies conducted to date with cannabinoids (e.g., as antiemetics or painkillers) in cancer patients.
10. Line 624. I of course agree with that cautionary conclusion. But would it also be possible that patients using cannabis were precisely those with a more complex symptomatology and worse overall health status/prognosis?
11. Line 703. Two very large cohort studies could be added: PMID 34940961 (Canada) and PMID 35223923 (Israel).
12. Line 705. PMID 36750134 could also be acknowledged.
13. Lines 714-on. In my opinion, the “Conclusions and Future Directions” section could be reinforced with some further speculation. For example, cancer is a very heterogeneous disease, so may anything be inferred from the current literature as to which precise cancer (sub)types/stages would be more likely to respond positively to cannabis products? Overall, could the authors provide some ideas on the key clinical studies that, in their expert opinion, should be conducted in the future for the field to move forward: which study designs, which (sub)types of cancers, which stages, in combination with which standard or experimental palliative/antineoplastic therapies, which cannabinoid types/doses, which routes of administration, etc.?
Reviewer 2 Report
The authors present a comprehensive review of the use and possible advantages/disadvantages of cannabis products in patients with cancer. I have the following remarks:
1. Line 83: I would suggest “to differentiate them from endocannabinoids …” to make the sentence clearer.
2. Line 229: Fungal infections in patients inhaling street cannabis is real problem, there is literature about it that could be cited, e.g. 10.3201/eid2606.191570. This is especially the case in immunocompromised patients.
3. Line 294: I suggest to present the chemical structures of THC and CBD. The metabolism of the compounds is easier to show and to gather with the structures.
4. Line 294: Is it Vd or Vd/F (apparent Vd)?
5. Line 327: Why is the half-life of THC/CBD longer in heavy users?
6. Line 534: I believe that the effect of THC/CBD on displacement of other drugs from the protein binding is low regarding the small dose and small concentrations reached with THC/CBD. The opposite may be true that THC/CBD are displaced from protein binding.
7. Line 555: Since most effects of cytotoxic drugs (the “old” anticancer drugs) are dose-dependent, a decrease in their metabolism would increase their toxicity if the dose is not reduced. If the dose is reduced to achieve the “normal” cytotoxic effect, I would expect similar adverse reactions (no reduction).
8. Line 670: A possible interaction may be with stimulant laxatives (although stimulant laxatives mainly stimulate the colon and not the small intestine where THC/CBD are absorbed). Since osmotic laxatives such as lactulose and macrogol act slowly, I see no interaction potential with them.
9. Line 683: Not thyroid cancer but parathyroid carcinoma and hyperparathyroidism for other reasons. Stimulates Ca-receptors, effect on phosphate excretion only secondary due to lowering of parathyroid hormone.
